# Characterization of Antibiotic-Resistance Antarctic *Pseudomonas* That Produce Bacteriocin-like Compounds

**DOI:** 10.3390/microorganisms12030530

**Published:** 2024-03-06

**Authors:** Nancy Calisto, Laura Navarro, Cristian Iribarren, Paz Orellana, Claudio Gómez, Lorena Salazar, Ana Gutiérrez, Carlos Aranda, Alex R. González, Mario Tello, Piedad Cortés-Cortés, Manuel Gidekel, Gino Corsini

**Affiliations:** 1Centro de Investigación y Monitoreo Ambiental Antártico (CIMAA), Departamento de Ingeniería Química, Universidad de Magallanes, Punta Arenas 6200000, Chile; nancy.calisto@umag.cl (N.C.); claudio.gomez@umag.cl (C.G.); 2Instituto de Ciencias Biomédicas, Facultad de Ciencias de la Salud, Universidad Autónoma de Chile, Santiago 8320000, Chile; laura.navarro@uautonoma.cl (L.N.); cristian.iribarren@uautonoma.cl (C.I.); paz.orellana@uautonoma.cl (P.O.); lorena.salazar@uautonoma.cl (L.S.); ana.gutierrez@uautonoma.cl (A.G.); mgidekel@gmail.com (M.G.); 3Departamento de Producción Agropecuaria, Facultad de Ciencias Agropecuarias y Medioambiente, Universidad de La Frontera, Temuco 4780000, Chile; 4Laboratorio de Microscopía Avanzada, Departamento de Ciencias Biológicas y Biodiversidad, Universidad de Los Lagos, Osorno 5290000, Chile; caranda@ulagos.cl; 5Laboratorio de Microbiología Ambiental y Extremófilos, Departamento de Ciencias Biológicas y Biodiversidad, Universidad de Los Lagos, Osorno 5290000, Chile; alex.gonzalez@ulagos.cl; 6Laboratorio de Metagenómica Bacteriana, Departamento de Biología, Facultad de Química y Biología, Universidad de Santiago de Chile, Santiago 8320000, Chile; mario.tello@usach.cl; 7Dirección de Desarrollo y Transferencia, Universidad de Las Américas, Santiago 8320000, Chile; pcortes@udla.cl

**Keywords:** bacteriocin, *Pseudomonas*, Antarctic bacteria, multi-drug resistance, pathogenic bacteria

## Abstract

In this study, bacterial isolates C1-4-7, D2-4-6, and M1-4-11 from Antarctic soil were phenotypically and genotypically characterized, and their antibacterial spectrum and that of cell-free culture supernatant were investigated. Finally, the effect of temperature and culture medium on the production of antimicrobial compounds was investigated. The three bacteria were identified as different strains of the genus *Pseudomonas*. The three bacteria were multi-drug resistant to antibiotics. They exhibited different patterns of growth inhibition of pathogenic bacteria. M1-4-11 was remarkable for inhibiting the entire set of pathogenic bacteria tested. All three bacteria demonstrated optimal production of antimicrobial compounds at 15 °C and 18 °C. Among the culture media studied, Nutrient broth would be the most suitable to promote the production of antimicrobial compounds. The thermostability exhibited by the antimicrobial molecules secreted, their size of less than 10 kDa, and their protein nature would indicate that these molecules are bacteriocin-like compounds.

## 1. Introduction

Since the discovery of penicillin in 1928, antibiotics, natural products produced by microorganisms capable of preventing the growth of bacteria and thus curing infectious diseases, have saved millions of lives. In the 1940s, for the first time, life-threatening diseases such as pneumonia or tuberculosis could be controlled and cured with antibiotics such as penicillin and streptomycin, respectively [1]. However, the inappropriate and excessive use of these drugs, both in the health sector and in the food industry, has caused a rapid spread of the phenomenon of antimicrobial resistance and the loss of efficacy of antibiotics in the treatment of some infections [2,3,4]. It has been projected that by 2050, infectious diseases could become the leading cause of death globally, with 10 million deaths each year, if antimicrobial resistance is not adequately addressed [2,5]. This complex situation has generated an urgent need to establish mechanisms to develop new and/or improved strategies to solve the increasing number of infections caused by bacteria resistant to antibiotics. Currently, important efforts are being made to achieve an appropriate use of existing antibiotics and promote the development of new drugs that can be used in clinical settings [5].

Collectively, microorganisms are capable of producing a wide variety of secondary metabolites, and the pharmaceutical industry has developed a few of these for therapeutic use [5,6,7]. These are important metabolites such as acids, alcohols, diacetals, and various antimicrobial compounds [3,8,9,10,11,12].

Bioprospecting of natural environments has led to the identification and commercialization of current antibiotics and continues to reveal new biomolecules with therapeutic potential. Currently, natural products, predominantly from bacteria and fungi, remain the main sources studied to discover new antibiotics [2,13]. In this context, the exploration of unusual new habitats and environments has become important to discover new microorganisms that generate antimicrobial metabolites. Microorganisms living in unique or extreme environments such as mine lakes, hyper-arid deserts, and the Antarctic continent may be promising starting points for finding new antimicrobial compounds [2,10,14,15]. Among the extreme environments, Antarctica has proven to be an interesting source of microbial species with great potential for various applications, including biomedicine through the discovery of antimicrobial compounds [13,16,17].

The antimicrobials produced by bacteria include bacteriocins, small ribosomally synthesized polypeptides, that kill or inhibit the growth of other bacteria [8,18].

Bacteriocins from many bacteria have been reported to be active against human and animal microbial pathogens without showing toxicity, which represents an advantage in the use of these types of compounds as antimicrobials [6]. Additionally, numerous bacteriocins have been identified and studied that show remarkable potential as food preservatives and therapeutic or biocontrol agents [4,19,20]. Therefore, some bacteriocins may be able to replace and/or complement current antibiotics.

In this work, three Antarctic soil bacteria (isolates C1-4-7, D2-4-6, and M1-4-11) with antimicrobial potential [21] were characterized phenotypically and genotypically, and both bacteria and cell-free culture supernatants were tested to determine their phylogenetic spectrum of antimicrobial activity. The effect of growth temperature and culture medium composition on antimicrobial production was also investigated.

## 2. Materials and Methods

### 2.1. Origin of the Antarctic Bacteria

The 3 Antarctic bacteria used in the present study were isolated from samples collected on two Antarctic islands belonging to the South Shetland Islands. Bacterial isolate designated C1-4-7 was isolated from a soil sample collected in Fildes Bay on King George Island (62°10′09″ S; 58°55′47″ W). Bacterial isolate D2-4-6 was isolated from a soil sample taken in Chile Bay on Greenwich Island (62°28′53″ S; 59°37′06″ W). Finally, the bacterial isolate designated M1-4-11 was isolated from the rhizosphere of *Deschampsia antarctica* Desv. collected near the Collins glacier on King George Island (62°10′07″ S; 58°51′06″ W). The Antarctic bacteria were obtained as described in Calisto et al., 2021 [21].

### 2.2. Morphological and Biochemical Characterization of C1-4-7, D2-4-6, and M1-4-11 Antarctic Bacteria

Morphological and biochemical tests were performed to identify Antarctic bacteria [22]. Bacterial isolates were studied for Gram stain, cell morphology, and sugar fermentation (glucose, fructose, sucrose, and lactose). Catalase activity was determined by the production of bubbles after the addition of a drop of 3% (*v*/*v*) hydrogen peroxide solution on the bacteria on a slide. The presence of oxidase activity was tested according to the manufacturer’s instructions for Mastdiscs^®^ ID oxidase (Mast Group Ltd., Bootle, Merseyside, UK). The presence of urease was tested using Urea agar (Becton, Dickinson, and Company, Sparks, MD, USA) according to the manufacturer’s instructions. Simmons citrate agar (AccumixTM Tulip Diagnostics Ltd., Goa, India) was used to test the ability of bacterial isolates to use citrate as the sole carbon source. Additionally, cetrimide agar was used as a medium to favor the growth of *Pseudomonas aeruginosa*, stimulate the production of its pigments and, in turn, inhibit the growth of other microorganisms.

Microscopy analysis was carried out using an optical light microscope (Motic BA 310 E) with 100X magnification. To perform the scanning electron microscopy (SEM) analyses, the bacterial samples were fixed in 2.5% glutaraldehyde in PBS. A 20 µL drop of the suspension was spread onto a carbon tape-covered coverslip attached to a 12 mm stub. The smear was dried at 37 °C and washed twice with distilled water to remove any remaining electrolyte crystals and glutaraldehyde. The salt-free dry smear was coated with a gold film using a Quorum Q150R ES metalizer. SEM images were obtained using a Carl Zeiss EVO 15 microscope with a tungsten filament under high vacuum and a secondary electron detector at 20 kV.

The enzyme profiles of Antarctic bacteria were determined using API ZYM strips (bioMerieux, Marcy l’Étoile, Lyon, France) according to the manufacturer’s instructions. The incubation time of the API strips was 24 h at 18 °C.

### 2.3. Antibiotic Susceptibility Profile of Antarctic Bacteria

The susceptibility of Antarctic bacteria to different antibiotics was determined using the disk diffusion method described previously [21]. For this, Mueller–Hinton agar plates and antibiotic disks belonging to different groups such as penicillins (ampicillin, amoxicillin-clavulanic acid, piperacillin-tazobactam), cephalosporins (cefuroxime, cefoxitin, cefotaxime, ceftazidime, cefepime, ceftriaxone), carbapenems (ertapenem, meropenem, imipenem), aminoglycosides (streptomycin, kanamycin, gentamicin, amikacin), quinolones (nalidixic acid, ciprofloxacin, levofloxacin), sulfonamides (sulfafurazole, cotrimoxazole), tetracycline, chloramphenicol, trimethoprim, and bacitracin were used. Bacteria were cultured at 18 °C for 48 h. From this, an inoculum was diluted in sterile distilled water until reaching a turbidity equivalent to 0.5 MacFarland. From this culture, an inoculum was seeded with a sterile swab on plates with Muller–Hinton agar, and on this inoculum, sensidiscs with the selected antibiotics were deposited. The plates were incubated at 18 °C for 24 h. *Escherichia coli* ATCC 25922 and *Staphylococcus aureus* ATCC 25923 were used as the control for the susceptibility tests. The diameter of the growth inhibition zone generated by each antibiotic was measured, and it was considered that an isolate was resistant to a given antibiotic when the diameter of the growth inhibition zone was less than or equal to 15 mm, intermediate susceptibility when the diameter was between 16 and 20 mm, and sensitive when the diameter of the growth inhibition zone was equal or greater than 21 mm [17].

### 2.4. Molecular Identification of Antarctic Bacteria

The genomic DNA of bacterial isolates was extracted with a Qiagen Genomic-tip 100/G kit (Qiagen, Courtaboeuf, Les Ulis, France) according to the manufacturer’s protocol. The integrity of the DNA obtained was evaluated by electrophoresis in 1% agarose gels. The DNA concentration and its purity were determined by its absorbance at 260 nm and with the 260/280 nm ratio, respectively, in the Tecan Infinite M200 PRO multiplate reader spectrophotometer.

Antarctic bacteria were identified through 16S rRNA sequence identity analysis. The polymerase chain reaction (PCR) was carried out with bacterial-specific 16S rRNA primers 27F (5′-AGAGTTTGATCMTGGCTCAG-3′) and 1492R (5′-CGGTTACCTTGTTACGACTT-3′) using genomic DNA as templates. The PCR conditions were initial denaturation at 95 °C for 3 min, followed by 35 cycles each of denaturation at 95 °C for 1 min, annealing at 55 °C for 1 min, and extension at 72 °C for 1.5 min, and a final extension at 72 °C for 5 min. The PCR products were separated on a 1% agarose gel in 1× TAE buffer, stained with GelRed^TM^ (Biotium Inc., Fremont, CA, USA), and visualized with UV light using a transilluminator. The PCR products were then purified with a GEL/PCR Purification mini-Kit (Favorgen Biotech Corp., Ping-Tung, Taiwan) and sequenced at Austral-Omics of the Universidad Austral de Chile, Valdivia, Chile.

BLAST analysis of 16S rRNA gene sequences was performed using the NCBI nucleotide database to identify similar sequences, and a maximum likelihood phylogenetic tree was used to identify the closest genus/species using IQTREE v.1.6.1 [23] and visualized with iTol v.4 [24]. A phylogenetic tree was created to determine the taxonomic status of Antarctic bacteria from samples collected from Antarctic soil.

### 2.5. Genotyping of Antarctic Bacteria by Repetitive Elements PCR Amplification

Using genomic DNA obtained from Antarctic bacteria, the sequences enterobacterial repetitive intergenic consensus (ERIC) and interspersed repetitive DNA sequence (BOX) were amplified using PCR to obtain genomic “fingerprint” patterns [25]. ERIC 1R (5′-ATGTAAGCTCCTGGGGATTCAC-3′) and ERIC 2 (5′-AAGTAAGTGACTGGGGTGAGCG-3′) primers were used to perform the ERIC-PCR. The PCR reaction conditions were initial denaturation at 94 °C for 5 min, followed by 35 cycles each of denaturation at 94 °C for 1 min, annealing at 48 °C for 1 min, and extension at 72 °C for 2 min, and a final extension at 72 °C for 5 min. PCR amplifications with BOX primers included only a single primer BOXA1R (5′-CTACGGCAAGGCGACGCTGACG-3′). PCR reaction conditions were an initial denaturation step at 95 °C for 7 min, followed by 30 cycles of denaturation at 90 °C for 30 s, annealing at 52 °C for 1 min, and extension at 65 °C for 8 min with a single final extension at 65 °C for 16 min.

The PCR products were separated on a 1.5% agarose gel in 1× TAE buffer at 50 V for 90 min, stained with GelRed^TM^ (Biotium Inc., Fremont, CA, USA), and visualized with UV light using a transilluminator.

### 2.6. Screening for Antimicrobial Activity of the Antarctic Bacteria and Their Cell-Free Culture Supernatants

Antimicrobial compound production by C1-4-7, D2-4-6, and M1-4-11 Antarctic bacteria was evaluated by plate activity assays as described previously [21], where it was detected whether the bacterial isolates secrete compounds that have the capacity to inhibit the growth of 18 pathogenic bacteria. The set of reference pathogenic bacteria studied includes Gram-positive bacteria of *Bacillus*, *Enterococcus*, *Micrococcus*, *Staphylococcus*, and *Streptococcus* genera, and Gram-negative bacteria of *Citrobacter*, *Enterobacter*, *Escherichia*, *Klebsiella*, *Proteus*, *Pseudomonas*, *Salmonella*, and *Vibrio* genera. Briefly, suspensions of overnight-grown pathogenic bacteria cultures were prepared, and an inoculum was seeded with a sterile swab on Muller–Hinton agar plates. Then, an inoculum of 5 µL of Antarctic bacteria, grown at 18 °C for 48 h, was deposited into the indicator bacteria lawn. Plates were incubated at 18 °C for 24 h for adequate growth of Antarctic bacterial isolates and then incubated at 37 °C for growth of indicator pathogenic bacteria. The result of the antagonistic activity was determined by the presence or absence of a growth inhibition zone in the lawn of each pathogenic bacteria analyzed.

The antibacterial activity of the cell-free culture supernatant was tested against the same set of pathogenic bacteria. The Antarctic bacteria were incubated at 18 °C for 60 h. The culture was centrifuged at 10,000 rpm for 20 min. The culture supernatant was concentrated 20-fold by evaporation at 80 °C. The antimicrobial activity of this concentrate was analyzed by the antimicrobial activity plate assay [26]. The culture plates were incubated at 37 °C for 24 h, and the growth inhibition zone in the lawn of each pathogenic bacteria was analyzed. Aliquots of 5 µL of Nut 1/3 broth concentrated 20-fold at 80 °C and 5 µL of *E. coli* growth supernatant concentrated 20-fold at 80 °C were used as negative controls. The antibiotic ampicillin (10 µg) was used as a positive control.

### 2.7. Exclusion Membrane Assay

To determine the approximate size of the molecules secreted by Antarctic bacteria that have antimicrobial activity, an antimicrobial activity plate assay [26] was used. On this occasion, the Antarctic bacteria and the cell-free concentrated culture supernatant were inoculated on a cellulose membrane with a 10 kDa exclusion pore (Spectra/Por^®^ 7 MWCO 10,000), previously deposited on a lawn of indicator pathogenic bacteria. The experiments were carried out following the same indications as these for the screening for antimicrobial activity of Antarctic bacteria.

### 2.8. Sensitivity of the Antimicrobial Agents to Proteinase K Enzyme

The proteinase K enzyme (Promega Corporation, Madison, WI, USA) was used to determine whether the inhibitory compounds produced by the Antarctic bacteria were proteinaceous [27]. The enzyme was prepared at a concentration of 25 mg/mL, according to the manufacturer’s instructions. The 20-fold concentrated supernatants were incubated for 2 h at 37 °C with proteinase K at a final concentration of 10 mg/mL. The antimicrobial activity of proteinase K-treated concentrated supernatant was analyzed by the antimicrobial activity plate assay [26]. Briefly, 5 µL of the concentrated supernatants treated with proteinase K were inoculated onto a lawn of reference pathogenic bacteria (*E. coli* and *S. aureus*). Culture plates were incubated at 37 °C for 24 h. The growth inhibition zone in the lawn of the reference bacteria was measured and compared with the growth inhibition zone produced by the concentrated supernatant without enzyme treatment. A 20-fold concentrated Nut 1/3 broth inoculated with nisin (a proteinase K-sensitive bacteriocin) was treated in the same way as the Antarctic bacterial concentrates and served as a positive control to assess proteinase K proteolytic activity.

### 2.9. Effect of Temperature and Culture Medium on the Growth and Bacteriocin-like Production of the Antarctic Bacteria

The ability of C1-4-7, D2-4-6, and M1-4-11 Antarctic bacteria to grow at different temperatures (4 °C, 10 °C, 15 °C, 18 °C, and 25 °C) was verified using nutrient broth diluted to one-third (Nut 1/3: peptone 1.67 g/L; meat extract 1 g/L; pH: 6.9). Bacterial growths were carried out simultaneously at all temperatures and without shaking.

The growth of bacterial isolates at 18 °C in different culture broths was analyzed. The following culture media were used: Luria Bertani (LB: tryptone 10 g/L; yeast extract 5 g/L; NaCl 10 g/L; pH: 7.1); LB diluted to a third (LB 1/3); nutrient (Nut: peptone 5 g/L; meat extract 3 g/L; pH: 6.9); nut diluted to a third (Nut 1/3); soy trypticase (TSB: casein digest 15 g/L; peptic digest of soy flour 5 g/L; glucose 2.5 g/L; NaCl 5 g/L; pH: 7.1) and minimal medium M63·supplemented with 0.2% glucose (M63 + 0.2%G: KH_2_PO_4_ 13.6 g/L; (NH_4_)_2_SO_4_ 2 g/L; FeSO_4_·7 H_2_O 0.5 mg/L; MgSO_4_·7 H_2_O 0.24 g/L; glucose 2 g/L; pH: 6.8). No shaking was used. Bacterial growth was monitored at 600 nm using a SPECTRO UV-11 spectrophotometer with 1 cm optical path plastic cuvettes.

After studying the effect of temperature and culture medium on the growth of Antarctic bacteria C1-4-7, D2-4-6, and M1-4-11, it was determined whether these variables affected the production of secreted antimicrobial compounds. For this purpose, the Antarctic bacteria were grown at different temperatures (4 °C, 10 °C, 15 °C, 18 °C, and 25 °C) and in different culture media (LB, LB 1/3, Nut, Nut 1/3, and TSB); then, the antimicrobial activity plate assay was performed, according to the methodology indicated above, and the zone of inhibition produced by the concentrated supernatant of each bacterium on *E. coli* and *S. aureus* was measured.

To relate bacterial growth to the production of antimicrobial compounds, the antimicrobial activity was expressed as arbitrary units per milliliter of 20-fold concentrated supernatant (AU/mL). AU was defined as the reciprocal of the highest two-fold dilution exhibiting a clear zone of inhibition on the indicator lawn.

### 2.10. Statistical Analysis

One-way analysis of variance (ANOVA) and Tukey’s non-parametric test for multiple comparisons were carried out using GraphPad Prism version 9.3.1 software. Differences between samples were considered statistically significant if the *p*-value was less than 0.05.

## 3. Results

### 3.1. Morphological and Biochemical Characterization of Antarctic Bacteria

The C1-4-7, D2-4-6, and M1-4-11 Antarctic bacteria were morphologically characterized as round, shiny colonies with regular borders and later by Gram staining under optical microscopy. It was observed that C1-4-7, D2-4-6, and M1-4-11 correspond to short Gram-negative bacilli. Additionally, the SEM images showed that isolates C1-4-7 ranged in size from 0.47 to 0.64 µm wide by 1.76–2.50 µm long, isolates D2-4-6 ranged from 0.53 to 0.68 µm wide by 1.03–2.57 µm, and finally isolates M1-4-11 ranged from 0.39 to 0.67 µm wide by 1.22–2.54 µm (Figure 1).

For the metabolic characterization of the Antarctic bacteria, different biochemical tests were carried out, and the results of these tests are presented in Table 1. The three Antarctic bacteria were positive for the enzyme catalase, oxidase, and urease tests. Additionally, the Antarctic bacteria presented a non-fermentative metabolism and the ability to grow in Simmons citrate, which is a medium that contains sodium citrate and inorganic ammonium salts as the only sources of carbon and nitrogen, respectively. Finally, the three bacteria were able to grow in cetrimide, which is a medium for selective growth of *Pseudomonas aeruginosa* and other species of the genus *Pseudomonas*. However, when the isolates were grown on cetrimide agar, no production of characteristic *Pseudomonas aeruginosa* pigments was observed (Appendix A).

The enzyme activity profile of Antarctic bacteria was determined using the commercial enzyme detection system API-ZYM (bioMerieux, Marcy l’Étoile, France). This colorimetric system allows, through color changes and comparison with controls, to detect the presence or absence of enzymatic activity. The enzymatic activity profiles of isolates C1-4-7, D2-4-6, and M1-4-11 are presented in Figure 2. Additionally, the enzymatic profile of *Pseudomonas aeruginosa* O400 used as a reference is included. The results show that the three Antarctic bacteria studied have detectable levels of the enzymes acid phosphatase, esterase (C4), and esterase lipase (C8) that participate in lipid metabolism. Additionally, in the three Antarctic bacteria, the enzyme leucine arylamidase (amino-peptidase) related to protein metabolism and the hydrolase naphthol-AS-BI-phosphohydrolase were detected. On the other hand, only C1-4-7 and M1-4-11 bacteria had detectable levels of alkaline phosphatase, and only isolate C1-4-7 had positive levels of valine arylamidase.

The Antarctic soil isolates all possessed some of the biochemical characteristics typical of *Pseudomonas* species.

### 3.2. Molecular Identification of Antarctic Bacteria

For molecular identification of the C1-4-7, D2-4-6, and M1-4-11 Antarctic bacteria, 16S rRNA sequence analysis of Antarctic bacteria C1-4-7, D2-4-6, and M1-4-11 was performed by phylogeny (Figure 3). The rooted tree shows the 16S rRNA sequence of the Antarctic bacteria clusters with the homologous sequences of microorganisms belonging to the genus *Pseudomonas*. As shown in Table 2, the Antarctic bacteria studied showed between 99.93% and 100% identify with bacteria of the genus *Pseudomonas* when BLAST analysis of 16S rRNA gene sequences was performed. This result, together with the results obtained in the morphological characterization of the Antarctic bacteria, allows us to classify the isolates as members of the genus *Pseudomonas*. Further characterization and comparison using the complete sequence of the bacterial genome could allow in the future to identify the species to which the Antarctic bacteria studied in this work belong.

### 3.3. Genotyping of Antarctic Bacteria by Detection of Repetitive Elements Amplified by PCR

When performing genomic identification using ERIC-PCR (Figure 4A) and BOX-PCR (Figure 4B) of *Pseudomonas* sp. C1-4-7, D2-4-6, and M1-4-11, the generation of different amplicon patterns was observed for each bacterium studied. PCR with ERIC and BOX primers allowed the visualization of fragments of different lengths that produced different fingerprints for *Pseudomonas* sp. C1-4-7, D2-4-6, and M1-4-11, indicating that they are different bacteria and confirming their phylogeny and enzyme profile.

### 3.4. Antibiotic Susceptibility Profile of Antarctic Pseudomonas *sp.* C1-4-7, D2-4-6, and M1-4-11

Table 3 shows the susceptibility of the isolates to antimicrobial agents. The three bacteria studied are multi-resistant since they present resistance to three or more groups of antibiotics.

The Antarctic bacteria (soil isolates) showed resistance or intermediate susceptibility to 16 of the 24 antibiotics studied. Collectively, they showed resistance to at least one antibiotic from each group tested.

### 3.5. Effect of Temperature and Culture Medium on the Growth of the Antarctic Bacteria C1-4-7, D2-4-6, and M1-4-11

Growth curves of the three isolates cultured in Nut 1/3 broth are shown in Figure 5. When analyzing the growth curves, it can be observed that as the temperature increases, the growth rate of the Antarctic bacteria increases; however, similar optical density (OD) values were reached at 150 h of growth, except for the curves observed at 4 °C.

The growth of *Pseudomonas* sp. C1-4-7, D2-4-6, and M1-4-11 in different culture broths (Nut, Nut 1/3, LB, LB 1/3, TBS and M63 + 0.2%G) was analyzed. The growth curves obtained are shown in Figure 5. Bacterial growth is very slight for *Pseudomonas* sp. M1-4-11 and D2-4-6 and practically undetectable for *Pseudomonas* sp. C1-4-7 when the M63 + 0.2%G broth is used for bacterial growth.

### 3.6. Screening for Antimicrobial Activity of the Antarctic Bacteria C1-4-7, D2-4-6, and M1-4-11 and Their Cell-Free Culture Supernatants Concentrated at 80 °C

The ability of the *Pseudomonas* sp. C1-4-7, D2-4-6, and M1-4-11 to inhibit the growth of a set of 18 human pathogenic bacteria was analyzed (Table 4; Appendix A). The three strains of Antarctic *Pseudomonas* presented different patterns of growth inhibition of the human pathogenic bacteria studied. The *Pseudomonas* sp. C1-4-7 secreted antimicrobial compounds with the ability to inhibit the growth of mainly Gram-negative pathogenic bacteria (*Salmonella enterica* serotype *Enteritidis*, *Klebsiella oxytocic*, *Vibrio parahaemolyticus* VpKX, *Klebsiella pneumoniae*, *Escherichia coli* XL-1 Blue, *Enterobacter aerogenes*). *Pseudomonas* sp. D2-4-6 secreted compounds that inhibit the growth of only Gram-positive pathogenic bacteria (*Staphylococcus saprophyticus*, *Micrococcus* sp., *Staphylococcus aureus*, *Streptococcus agalactiae*). Finally, the *Pseudomonas* sp. M1-4-11 secreted antimicrobial compounds capable of inhibiting all pathogenic bacteria studied.

Additionally, to determine if the compounds with antimicrobial activity secreted by the Antarctic bacteria are thermostable, the bacteria-free culture supernatants were concentrated at 80 °C. Then, the ability of these concentrates to inhibit the growth of pathogenic bacteria was determined (Table 4; Appendix A). It can be noted that both the *Pseudomonas* sp. M1-4-11 and its concentrated cell-free supernatant can inhibit the entire set of pathogenic bacteria studied. In the case of *Pseudomonas* sp. D2-4-6, the concentrated cell-free supernatant was observed to inhibit the growth of a greater number of pathogenic bacteria than the bacterial isolate.

### 3.7. Exclusion Membrane Assay

Antimicrobial activity plate assays were performed using 10 kDa pore size exclusion membranes located between the reference pathogenic bacteria and the Antarctic bacteria or cell-free culture supernatant concentrate. It was observed that the compounds with antimicrobial activity were able to cross the exclusion membrane and inhibit the growth of the reference pathogenic bacteria (Appendix A). This result indicates that the Antarctic bacteria studied secrete antimicrobial compounds that are less than 10 kDa in size.

### 3.8. Sensitivity of the Antimicrobial Agents to Proteinase K Enzyme

The antimicrobial compounds produced by *Pseudomonas* sp. C1-4-7, D2-4-6, and M1-4-11 were sensitive to Proteinase K treatment. After treating the cell-free concentrated supernatants with Proteinase K, none of them showed antimicrobial activity on the reference pathogenic bacteria (*E. coli* and *S. aureus*) (Appendix A). This result indicated that the antimicrobial compounds produced by these *Pseudomonas* sp. from Antarctica are proteinaceous.

### 3.9. Effect of Temperature and Culture Medium on the Bacteriocin-like Production of the Antarctic Bacteria C1-4-7, D2-4-6, and M1-4-11

To determine at what stage of growth of the *Pseudomonas* sp. C1-4-7, D2-4-6 and M1-4-11 bacteria the antimicrobial compounds are produced, the growth curves of the bacteria were compared with the antimicrobial activity of concentrated supernatants obtained at various time points during bacterial growth (0, 18, 42, 66, and 90 h). A time of 90 h was chosen since the three Antarctic bacteria reached the stationary phase at that time. Figure 6 shows the results obtained when the experiments were carried out at 18 °C and using LB as a culture medium. *S. aureus* was used as an indicator of pathogenic bacteria for *Pseudomonas* sp. D2-4-6 and M1-4-7 and *E. coli* for *Pseudomonas* sp. C1-4-7.

To determine whether the temperature and the culture broths used for the growth of Antarctic bacteria affect the production of bacteriocins, the growth inhibition zone of reference pathogenic bacteria produced by *Pseudomonas* sp. C1-4-7, D2-4-6, and M1-4-11 at different temperatures and using different culture broths were analyzed.

Figure 7 shows the results obtained for the different growth temperatures studied using nut 1/3 broth. It can be observed for *Pseudomonas* sp. C1-4-7 that the inhibition zone is only significantly lower when the bacterium grows at 25 °C. For *Pseudomonas* sp. D2-4-6, no growth inhibition zone was observed for *S. aureus* when the bacteria grew at 4 °C. At 18 °C, a significantly larger inhibition zone was obtained than that observed for the other temperatures studied. Finally, for *Pseudomonas* sp. M1-4-7, it was observed that at 10 °C and 15 °C, significantly larger inhibition zones were obtained for both *E. coli* and *S. aureus*.

Figure 7 shows the results obtained for the growth inhibition zone of reference pathogenic bacteria produced by Antarctic bacteria when grown in different culture broths at 18 °C. Figure 7 shows that for *Pseudomonas* sp. C1-4-7, significantly greater growth inhibition zones were obtained for *E. coli* when nut and LB 1/3 broths were used for bacterial growth. In the case of *Pseudomonas* sp. D2-4-7, the largest growth inhibition zones for *S. aureus* were obtained when the bacteria were grown in LB and TSB broths. In contrast to the other strains studied, *Pseudomonas* sp. D2-4-7 did not exhibit growth inhibition of *S. aureus* at 4 °C. Finally, for *Pseudomonas* sp. M1-4-11, the growth inhibition zone of *E. coli* was not modified depending on the culture broth studied.

## 4. Discussion

The data on Antarctic microorganisms that produce antimicrobial compounds are scarce compared to other widely studied environments [28,29]. Bacteria isolated from the Antarctic environment currently remain a potential and underexplored source of new antimicrobial compounds, which may be advantageous in food, therapeutic, health, and industrial applications in the future.

In the first part of this study, we worked on the identification and characterization of three bacteria isolated from Antarctic soil samples collected in sectors far from human influence. These bacteria were selected because they belong to different sampling sites and because they secrete antibacterial molecules with different growth inhibition patterns against a set of eight reference human pathogenic bacteria [21]. The results of the analysis of the 16S rDNA gene sequences confirmed that the Antarctic bacteria belong to the genus *Pseudomonas*. The ERIC-PCR and BOX-PCR genomic fingerprinting of *Pseudomonas* sp. C1-4-7, D2-4-6, and M1-4-11 indicated that the three bacteria are different from each other. The metabolic and enzymatic characteristics of the Antarctic bacterial isolates are like those reported by other authors for bacteria of the genus *Pseudomonas* isolated from soil samples [30,31,32].

*Pseudomonas* strains are found in large numbers in all the major natural environments (terrestrial, freshwater, and marine) and form intimate associations with plants and animals [33]. The population of such environments involves a struggle for living space and organic nutrients with many other microorganisms. *Pseudomonas* display a highly versatile metabolism, and several secondary metabolites affecting other bacteria and fungi that are attracted to common niches have been identified [34]. Different authors have reported the presence of this bacterial genus in Antarctica, both at the soil and sediment level [10,17,31,35,36] and associated with the growth of vascular plants such as *D. antarctica* [37].

The *Pseudomonas* sp. C1-4-7, D2-4-6, and M1-4-11 showed multi-drug resistance, which correlates with several studies that have reported the presence of resistance to different antibiotics in bacteria isolated from Antarctic soils [17,36,38,39,40]. Tomova et al. (2015) [17] suggest that the antibiotic multiresistance detected in their study could be associated with an anthropogenic impact on the Antarctic islands Deception and Galíndez. On the other hand, Marcoleta et al. (2022) [39] found a higher general resistance in bacteria from areas not intervened by human activities. In their study, they detected *Pseudomonas* resistant to 10 or more different antibiotics, a result similar to that observed in this study where bacteria were isolated from soil samples obtained in sectors without anthropogenic intervention. This supports our initial findings reported in Calisto et al. (2021) [21].

The results of the first part of this study suggest that these Antarctic bacteria are potential sources of genes encoding for both antimicrobial compounds and resistance to antibiotics. From an ecological point of view, these two capabilities probably provided the competitive advantage to these Antarctic bacteria to survive in the harsh environment [36,41], while on the other hand, these same characteristics may be of interest for biomedical applications.

The second part of this work aimed to analyze the antimicrobial activity of *Pseudomonas* sp. C1-4-7, D2-4-6, and M1-4-11 and their cell-free supernatants. It was observed that each bacterium studied showed different patterns of growth inhibition of the set of pathogenic bacteria tested. The growth inhibition of all pathogenic bacteria tested by *Pseudomonas* sp. M1-4-11 is remarkable. When the antimicrobial activity of the cell-free supernatant concentrated at 80 °C of *Pseudomonas* sp. M1-4-11 was analyzed, the same inhibition pattern observed for the bacterium was found. However, for the cell-free concentrated supernatant of *Pseudomonas* sp. C1-4-7, a different pattern of antimicrobial activity was observed. The concentrated supernatant was able to inhibit the growth of pathogenic bacteria that *Pseudomonas* sp. C1-4-7 could not inhibit, but this concentration lost the capacity to inhibit other pathogens that the bacteria could directly inhibit. Finally, the concentrated supernatant of *Pseudomonas* sp. D2-4-6 showed antimicrobial activity in a group of Gram-negative pathogenic bacteria on which bacteria did not present antimicrobial activity. The loss of antimicrobial activity of the concentrated supernatant in relation to the bacteria could be because the bacteria could secrete antimicrobial compounds that are not stable at high temperatures; therefore, they degrade when concentrating the supernatant at 80 °C. On the other hand, the increase in the antimicrobial activity of the concentrated supernatant with respect to the activity presented by the bacteria could be explained if it is considered that the bacteria can secrete small amounts of some thermostable antimicrobial compounds; therefore, the antimicrobial activity of these compounds is only evidenced by concentrating the supernatant 20 times.

Different authors have reported that Antarctic bacteria are a potential source of antimicrobial compounds, and bacteria with antimicrobial activity have been isolated from various Antarctic habitats, predominantly soils, sediments, and seawater [2,10,13,14,15,16,17,28,41,42,43]. Several authors have reported bacteria of the genus *Pseudomonas* isolated from Antarctic soils that can inhibit the growth of different groups of reference bacteria [10,17,36,43,44]. Wong et al. (2011) [36] identified five bacteria of the genus *Pseudomonas* that showed different patterns of growth inhibition of pathogenic bacteria. Among the 24 isolates obtained by Tomova et al. (2015) [17], the *Pseudomonas* sp. A1-1 exhibited the broadest inhibitory spectrum, being active against all reference bacteria and yeast cultures tested. On the other hand, Silva et al. (2018) [10] highlighted *Pseudomonas* sp. isolate 99 for showing a broad antimicrobial spectrum, in addition to antiproliferative and antiparasitic activity. Recently, Orellana et al. (2022) [43] reported 11 bacteria of the genus *Pseudomonas* isolated from the rhizosphere of *Deschampsia antarctica* Desv. with the ability to inhibit the growth of different human pathogenic bacteria. None of the works mentioned above report the antimicrobial capacity of the bacteria and their cell-free supernatant.

Despite the diversity of studies in the academic literature pointing to the commercial and therapeutic potential of Antarctic bacteria in the field of antimicrobials, in the study carried out by Silva et al. (2022) [29], which was based on a bibliographic search by patent documents, only four patent documents were found mentioning Antarctic bacteria, which were associated with five bacteria identified as *Janthinobacterium* sp., *Flavobacterium* sp., *Streptomyces radiopugnans*, *Streptomyces* sp., and *Bacillus* sp. This situation could be indicate that many of the studies that have reported Antarctic bacteria with antimicrobial activity have been limited only to identifying these bacteria in different environmental matrices but have not continued to study the characteristics of the compounds produced by these bacteria.

The last part of this study focused on performing an initial characterization of the antimicrobial compounds secreted by *Pseudomonas* sp. C1-4-7, D2-4-6, and M1-4-11 and analyzing how the temperature and culture media used for the growth of these Antarctic bacteria affect the production of antimicrobial compounds.

The results in Figure 6 and Figure 7 show that the growth of *Pseudomonas* sp. C1-4-7, D2-4-6, and M1-4-11 and the production of antimicrobial compounds are different for each of the bacteria under the same growth conditions. However, it could be indicated that the three bacteria have an optimal production of antimicrobial compounds between 15 °C and 18 °C. Among the culture media studied, nut would be the most suitable for favoring the production of antimicrobial compounds for *Pseudomonas* sp. C1-4-7, D2-4-6, and M1-4-11.

The thermostability exhibited by antimicrobial compounds secreted by *Pseudomonas* sp. C1-4-7, D2-4-6, and M1-4-11 when concentrating the cell-free supernatant at 80 °C, the size less than 10 kDa, and the protein nature of the antimicrobial molecules would indicate that these compounds are bacteriocins [20,45]. Bacteriocins are small peptides or polypeptides produced by Gram-positive and Gram-negative bacteria, ribosomally synthesized, and very heterogeneous regarding their size, structure, mechanisms of action, spectrum of activity, biochemical properties, and target cell receptors [46,47]. It is considered that the majority of bacteria generate at least one antimicrobial peptide for self-preservation and competitive advantages in their ecological niche [45].

In this work, we studied three *Pseudomonas* obtained from Antarctic soil. It was found that these three Pseudomonas were able to produce antimicrobial bacteriocin-like compounds, which have different patterns of inhibition of human pathogenic bacteria. It was also found that the production of bacteriocin-like compounds can be modified with temperature and with the culture mediums used for bacterial growth. These results indicate that bacteriocin-like compounds secreted by Antarctic bacteria C1-4-7, D2-4-6, and M1-4-11 could be promising new antimicrobial compounds with potential therapeutic and food preservation applications.

Among the future perspectives of this study, we propose to sequence the complete genomes to identify the genes responsible for encoding the bacteriocin-like compounds in each of the *Pseudomonas*. This could lead to their subsequent expression in a heterologous system to ultimately improve their production.

## Figures and Tables

**Figure 1 microorganisms-12-00530-f001:**
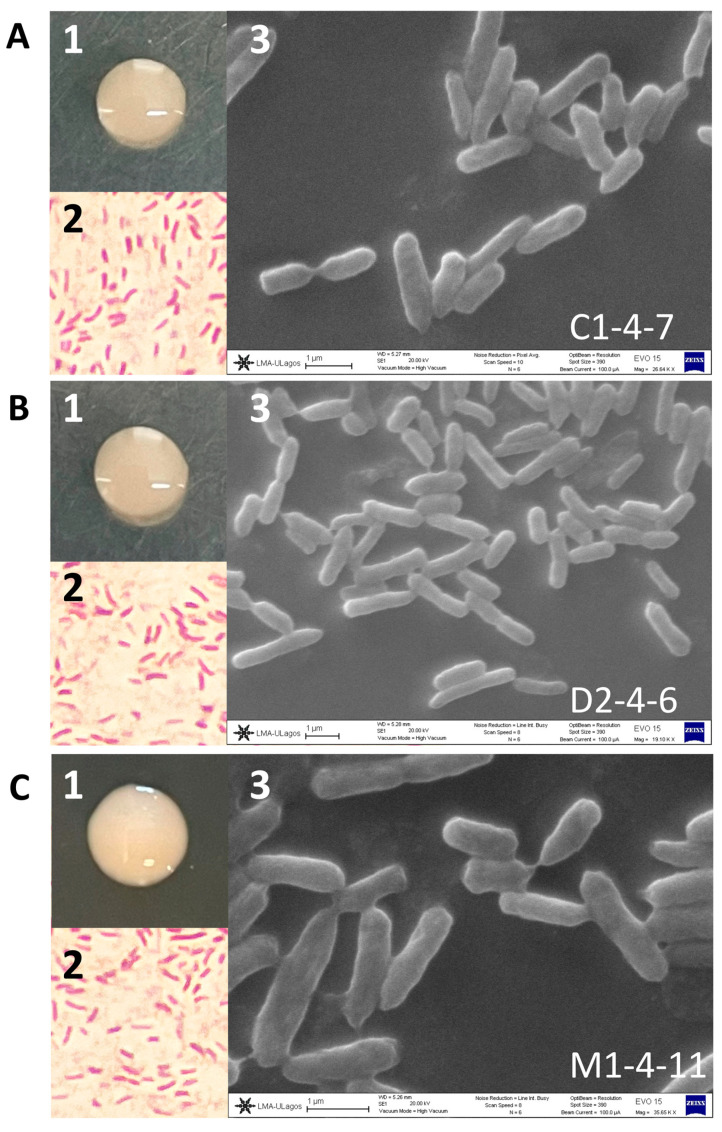
Morphology and microscopy of Antarctic soil isolates (**A**) C1-4-7, (**B**) D2-4-6, and (**C**) M1-4-11. (1) Bacterial colony. (2) Gram stain microscopy. (3) Electron microscopy.

**Figure 2 microorganisms-12-00530-f002:**
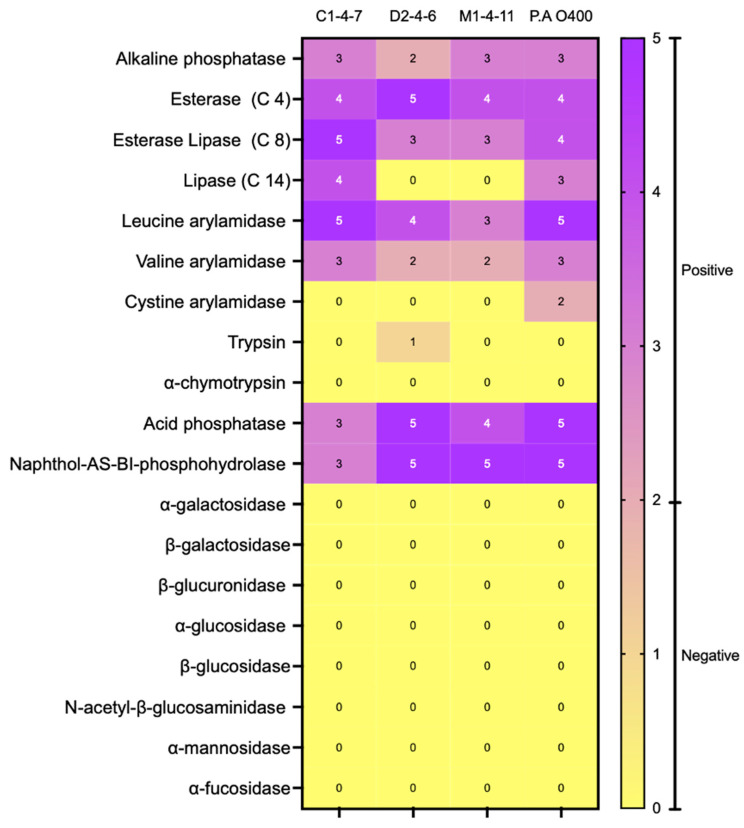
Enzyme activity assay (API-ZYM) of Antarctic bacteria C1-4-7, D2-4-6, and M1-4-11. According to the instructions indicated by the manufacturer, a negative result is obtained when the intensity of the observed color is between 0 and 2, while positive results correspond to values between 3 and 5.

**Figure 3 microorganisms-12-00530-f003:**
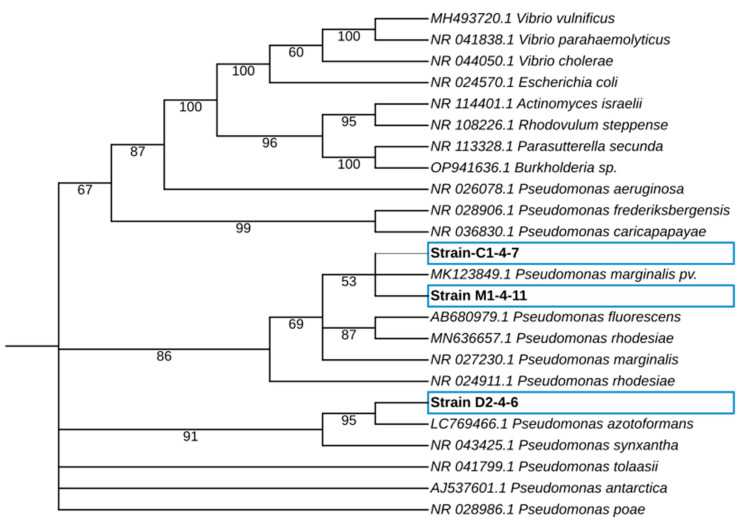
Maximum-likelihood phylogenetic tree-based 16S rRNA gene sequences of Antarctic bacteria C1-4-7, D2-4-6, and M1-4-11. The phylogenetic tree was obtained using the IQTREE v2.2.0 software, and the iTol program was used to visualize the phylogenetic tree. The percentage of bootstrap values is shown next to the branches based on 10,000 bootstrap replications.

**Figure 4 microorganisms-12-00530-f004:**
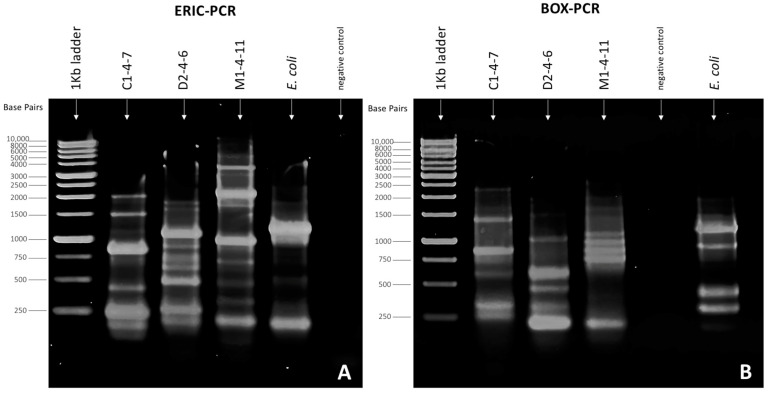
Fingerprinting patterns obtained for *Pseudomonas* sp. C1-4-7, D2-4-6, and M1-4-11 using (**A**) ERIC-PCR and (**B**) BOX-PCR. Electrophoresis conditions: 10 μL sample, 1.5% agarose gel, 50 Volt, and 90 min.

**Figure 5 microorganisms-12-00530-f005:**
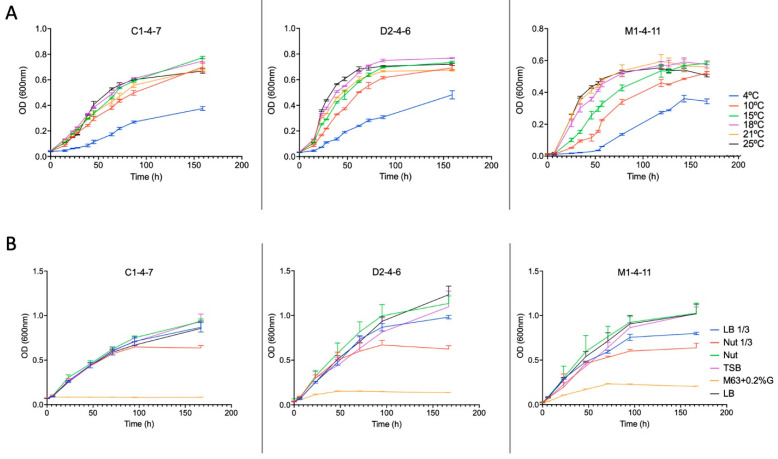
Growth curves of Antarctic bacteria C1-4-7, D2-4-6, and M1-4-11 (**A**) at different temperatures and (**B**) in different culture media at 18 °C.

**Figure 6 microorganisms-12-00530-f006:**
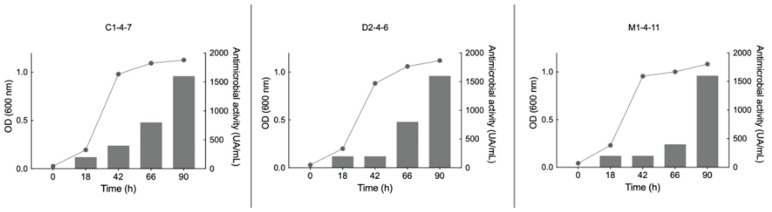
Relationship between growth curves (left axis line graph) and growth inhibition zone (right axis bar graph) produced by the concentrated supernatant of Antarctic bacteria C1-4-7, D2-4-6, and M1-4-11 on reference pathogenic bacteria. For C1-4-7, *E. coli* was used as a reference and for D1-4-6 and M1-4-11, *S. aureus* was used as a reference. UA/mL: arbitrary unit per milliliter.

**Figure 7 microorganisms-12-00530-f007:**
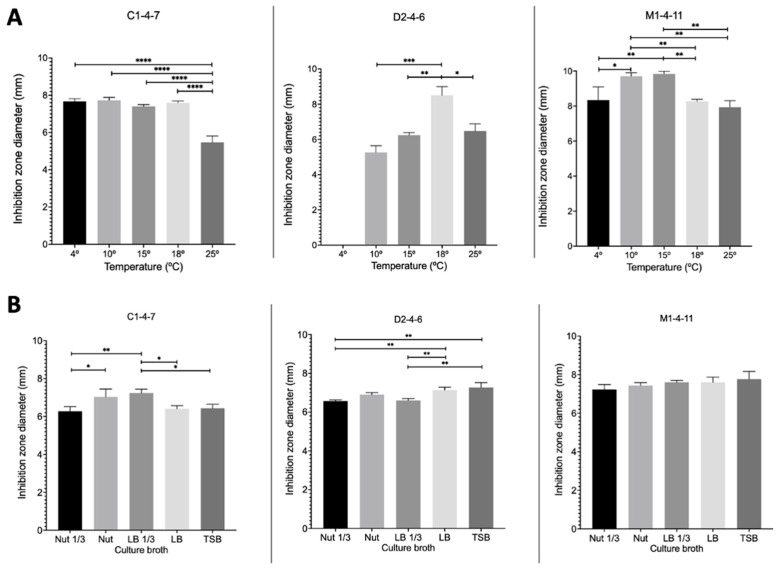
Inhibition zones at different temperatures (**A**) and in different culture broths (**B**) produced by *Pseudomonas* sp. C1-4-7, D2-4-6, and M1-4-11 on reference pathogenic bacteria. For C1-4-7, *E. coli* was used as a reference and for D1-4-6 and M1-4-11, *S. aureus* was used as a reference. **** *p* < 0.0001; *** *p* < 0.001; ** *p* < 0.01; * *p* < 0.05 (*n* = 3).

**Table 1 microorganisms-12-00530-t001:** Results of biochemical tests performed on the Antarctic bacteria C1-4-7, D2-4-6, and M1-4-11.

Biochemical Test	C1-4-7, D2-4-6, and M1-4-11
Catalase	+
Oxidase	+
Urease	+
Use of citrate as a carbon source	+
Growth on Cetrimide	+
Glucose Fermentation	−
Fructose Fermentation	−
Sucrose Fermentation	−
Lactose Fermentation	−

**Table 2 microorganisms-12-00530-t002:** Identification of Antarctic bacteria based on 16S ribosomal DNA sequencing and sequence analysis by BLAST alignment.

Antarctic Bacteria	Closest Related Species	Identity	Accession ID
C1-4-7 ID: OR839092	*Pseudomonas marginalis* pv. Marginalis	99.93%	MK123849.1
	*Pseudomonas fluorescens* NBRC 15840		AB680979.1
	*Pseudomonas rhodesiae* BDNA-E25		MN636657.1
	*Pseudomonas* sp. *G.S.34*		MT890191.1
D2-4-6 ID: OR839091	*Pseudomonas* sp. J380	100%	CP043060.1
	*Pseudomonas azotoformans* P20-L3		LC769466.1
	*Pseudomonas carnis* UCD_MED7		ON595689.1
	*Pseudomonas azotoformans* PCRB17a		ON564745.1
M1-4-11 ID: OR839093	*Pseudomonas marginalis* pv. Marginalis	99.93%	MK123849.1
	*Pseudomonas fluorescens* NBRC 15840		AB680979.1
	*Pseudomonas rhodesiae* BDNA-E25		MN636657.1
	*Pseudomonas* sp. W15Feb39		EU681022.1

**Table 3 microorganisms-12-00530-t003:** Antibiotic susceptibility profile of Antarctic bacteria C1-4-7, D2-4-6, and M1-4-11.

	Antibiotics	μg	C1-4-7	D2-4-6	M1-4-11
Penicillins	Ampicillin	10	R	R	R
	Amoxicillin-Clavulanic Acid	10–20	R	R	R
	Piperacillin-Tazobactam	10–100	S	S	S
Cephalosporins	Cefuroxime	30	R	R	R
	Ceftazidime	30	S	S	S
	Cefepime	30	S	I	I
	Cefotaxime	30	R	R	R
	Ceftriaxone	30	R	S	R
Aminoglycosides	Amikacin	30	S	S	S
	Gentamicin	10	S	S	S
	Kanamycin	30	S	S	S
	Streptomycin	10	I	R	S
Quinolones	Ciprofloxacin	5	S	S	S
	Nalidixic acid	30	S	I	S
	Levofloxacin	5	S	S	S
Sulfonamides	Sulfafurazole	300	R	R	I
	Cotrimoxazole	25	R	I	R
Carbapenems	Ertapenem	10	R	R	R
	Meropenem	10	S	R	S
	Imipenem	10	S	S	S
Others	Trimethoprim	5	R	R	R
	Chloramphenicol	30	R	R	R
	Tetracycline	30	S	I	S
	Bacitracin	0.04 U	R	R	R

S: sensitive (≥21 mm); I: intermediate (16–20 mm); R: resistant (≤15 mm).

**Table 4 microorganisms-12-00530-t004:** Antagonistic effect of Antarctic bacteria C1-4-7, D2-4-6, and M1-4-11 against bacterial pathogens.

Pathogenic Bacteria		C1-4-7		D2-4-6		M1-4-11	
		Bacteria	Cell-Free Supernatant *	Bacteria	Cell-Free Supernatant *	Bacteria	Cell-Free Supernatant *
Gram positive	*Staphylococcus saprophyticus*	−	−	+	+	+	+
	*Micrococcus* sp.	−	−	+	+	+	+
	*Bacillus cereus*	−	−	−	−	+	+
	*Staphylococcus aureus* ATCC 25923	−	−	+	+	+	+
	*Staphylococcus aureus* ATCC 6835	+	−	−	−	+	+
	*Staphylococcus epidermidis*	−	−	−	−	+	+
	*Enterococcus faecium*	−	−	−	−	+	+
	*Streptococcus agalactiae*	−	+	+	+	+	+
Gram negative	*Salmonella enterica* serotype Enteritidis	+	+	−	−	+	+
	*Klebsiella oxytocic*	+	−	−	−	+	+
	*Vibrio parahaemolyticus* VpKX	+	+	−	−	+	+
	*Klebsiella pneumoniae*	+	+	−	−	+	+
	*Proteus vulgaris*	−	−	−	+	+	+
	*Pseudomonas aeruginosa* O400	−	+	−	+	+	+
	*Escherichia coli* XL-1 Blue	+	+	−	+	+	+
	*Pseudomonas aeruginosa* PAO1	−	+	−	+	+	+
	*Enterobacter aerogenes*	+	−	−	−	+	+
	*Citrobacter freundii*	−	−	−	−	+	+

+: growth inhibition, −: no growth inhibition, * concentrated 20-fold at 80 °C.

## Data Availability

The study data are present in the main text, and for further inquiries please contact the corresponding author.

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
