# Peer review of "Characterization of Antibiotic-Resistance Antarctic Pseudomonas That Produce Bacteriocin-like Compounds"

_microorganisms, 2024, doi:10.3390/microorganisms12030530_

Round 1

Reviewer 1 Report

Comments and Suggestions for Authors

Comments on the Quality of English Language

Reviewer 2 Report

Comments and Suggestions for Authors

This manuscript provide the bacteriocin-like compounds secreted by Antarctic bacteria C1 -4-7, D2-4-6, and M1-4-11, which could be promising as new antimicrobial compounds to be used in biomedical applications or in the food field. There are some places can be improved.

1> Abstract Part: improve the two scentences in lines 26-27. "Multidrug" or multi-drug? it should be uniform.

2> Materials and Methods Part: add references for all the methods if necessary. Italy the name of bacteria in lines 134-135.

3> Results Part:  Line 356, in different culture broths at 18 ºC. Figure  6 was not clear, Please improve it.

4> Discussion Part: "The results of the first part of this study showed that" and others are the repeat of results. Please improve them.

Comments on the Quality of English Language

This manuscript provide the bacteriocin-like compounds secreted by Antarctic bacteria C1 -4-7, D2-4-6, and M1-4-11, which could be promising as new antimicrobial compounds to be used in biomedical applications or in the food field. There are some places can be improved.

1> Abstract Part: improve the two scentences in lines 26-27. "Multidrug" or multi-drug? it should be uniform.

2> Materials and Methods Part: add references for all the methods if necessary. Italy the name of bacteria in lines 134-135.

3> Results Part:  Line 356, in different culture broths at 18 ºC. Figure  6 was not clear, Please improve it.

4> Discussion Part: "The results of the first part of this study showed that" and others are the repeat of results. Please improve them.

Round 2

Reviewer 1 Report

Comments and Suggestions for Authors

The revised manuscript comprehensively addresses the points raised in the first review.

Still not convinced about the value of Table 1 as formatted - for example, since the three isolates have identical biochemical test results, there is no need for a separate column for each strain.  All three strain identifiers could be combined at the top of a single test results (+/-) column.

The photographs of indicator agar plates (in the Supplementary file) are helpful.  In Figure S3, presumably E. coli (EC) is the indicator pathogen in panel A, and S. aureus (SA) is the indicator pathogen in panels B and C.  If that's the case, then consider making this more obvious for the reader by spelling it out in the figure legend.

Comments on the Quality of English Language

Seems OK - the meaning is clear enough.

On line 489, consider: of against a set of 8 reference human pathogenic bacteria

Author Response

Thank you very much for taking the time to review this manuscript. Please find the detailed responses below and the corresponding corrections in track changes in the re-submitted files.

Still not convinced about the value of Table 1 as formatted - for example, since the three isolates have identical biochemical test results, there is no need for a separate column for each strain.  All three strain identifiers could be combined at the top of a single test results (+/-) column.

Answer: Accepted and modified. At the top of a single test result column, all three strain identifiers were combined.

The photographs of indicator agar plates (in the Supplementary file) are helpful.  In Figure S3, presumably E. coli (EC) is the indicator pathogen in panel A, and S. aureus (SA) is the indicator pathogen in panels B and C.  If that's the case, then consider making this more obvious for the reader by spelling it out in the figure legend.

Answer: Accepted and modified. In the legend to Figure S3, the following has been added: Indicator pathogen: A: E. coli (EC); B y C: S. aureus (SA).

Response to Comments on the Quality of English Language

On line 489, consider: of against a set of 8 reference human pathogenic bacteria.

Answer: Accepted and modified.